# Qualification of Hemophilia Treatment Centers to Enable Multi-Center Studies of Gene Expression Signatures in Blood Cells from Pediatric Patients

**DOI:** 10.3390/jcm12052080

**Published:** 2023-03-06

**Authors:** Birgit M. Reipert, Christoph J. Hofbauer, Bagirath Gangadharan, Verena Berg, Elizabeth Donnachie, Shannon Meeks, Maria Elisa Mancuso, Joel Bowen, Deborah L. Brown

**Affiliations:** 1Baxalta Innovations GmbH, Takeda Company, 1220 Vienna, Austria; 2Krems Bioanalytics, IMC University of Applied Sciences Krems, 3500 Krems an der Donau, Austria; 3College of Pharmacy and Health Sciences, University of Texas Health Science Center at Houston, Houston, TX 77030, USA; 4Aflac Cancer and Blood Disorders Center, Children’s Healthcare of Atlanta, Emory University, Atlanta, GA 30342, USA; 5IRCCS Humanitas Research Hospital, Rozzano, 20089 Milan, Italy; 6Fondazione IRCCS Ca’ Granda Ospedale Maggiore Policlinico, 20122 Milan, Italy; 7Indiana Hemophilia and Thrombosis Center Inc., Indianapolis, IN 46260, USA; 8Department of Pediatrics, University of Texas Health Science Center at Houston, Houston, TX 77030, USA

**Keywords:** hemophilia A, multi-center clinical research study, gene expression signatures in human PBMC, quality assurance program, qualification of operators in Hemophilia treatment centers

## Abstract

Hemophilia A is a rare congenital bleeding disorder caused by a deficiency of functionally active coagulation factor VIII (FVIII). Most patients with the severe form of the disease require FVIII replacement therapies, which are often associated with the development of neutralizing antibodies against FVIII. Why some patients develop neutralizing antibodies while others do not is not fully understood. Previously, we could demonstrate that the analysis of FVIII-induced gene expression signatures in peripheral blood mononuclear cells (PBMC) obtained from patients exposed to FVIII replacement therapies provides novel insights into underlying immune mechanisms regulating the development of different populations of FVIII-specific antibodies. The aim of the study described in this manuscript was the development of training and qualification test procedures to enable local operators in different European and US clinical Hemophilia Treatment Centers (HTC) to produce reliable and valid data for antigen-induced gene expression signatures in PBMC obtained from small blood volumes. For this purpose, we used the model antigen Cytomegalovirus (CMV) phosphoprotein (pp) 65. We trained and qualified 39 local HTC operators from 15 clinical sites in Europe and the US, of whom 31 operators passed the qualification at first attempt, and eight operators passed at the second attempt.

## 1. Introduction

Hemophilia A is a rare congenital bleeding disorder caused by the deficiency of biologically active coagulation factor VIII (FVIII). Most patients are treated with FVIII replacement therapies. Approximately 30% of previously untreated patients (PUP) with severe hemophilia A develop neutralizing antibodies (FVIII inhibitors) following replacement therapies, which is their main complication in children with severe hemophilia A [1,2,3]. Patients with FVIII inhibitors have substantially increased morbidity and increased cost of care [4,5,6]. On average, these patients develop FVIII inhibitors after 5–15 exposure days to FVIII concentrates [3,4,5,6,7].

Several risk factors for inhibitor development have been identified over recent decades, but why some patients develop FVIII inhibitors while others do not is not fully understood. Several clinical studies have shown that the development of FVIII inhibitors represents only a fraction of the overall antibody response against FVIII observed in patients who receive replacement therapies with FVIII products [8,9,10,11]. Longitudinal antibody data from the Hemophilia Inhibitor PUP Study (HIPS; www.clinicaltrials.gov, #NCT01652027) revealed four subgroups of patients who developed distinct signatures of FVIII inhibitors and FVIII-binding antibodies in response to the first 50 exposure days (ED) to a single recombinant FVIII product [11]. In order to explain the underlying immune mechanisms that are responsible for the lack of detectable antibody responses in some patients and for the generation of high- or low-affinity non-neutralizing or neutralizing antibodies against FVIII in other patients, a good understanding of mechanisms associated with these different antibody responses is required. 

A thorough understanding of the role of cellular immunity, including T cells and innate immune cells, in the regulation of the different antibody responses against FVIII would be particularly relevant. However, monitoring FVIII-specific T cells and other immune cells during a prospective multicenter clinical study in pediatric patients is very challenging and restricted to circulating immune cells detectable in peripheral blood. Moreover, the generation of longitudinal data using circulating immune cells in a hemophilia A PUP trial requires technologies that can be easily adapted to the small volumes of peripheral blood obtainable from children often younger than 1 year of age (e.g., 3–4 mL) in a prospective international multicenter setting 

Previously, Hofbauer et al. presented proof-of-principle data indicating that the presence of high-affinity FVIII-specific IgG4 antibodies and FVIII inhibitors in patients with severe hemophilia A correlated with the upregulation of pro-inflammatory gene expression signatures in PBMC after short-term in vitro re-stimulation with FVIII [12,13,14]. FVIII-induced pro-inflammatory signatures disappeared after successful Immune Tolerance Induction (ITI) therapy [12,13,14]. Recently, Karim et al. published similar findings indicating the upregulation of distinct pro-inflammatory transcriptome signatures in PBMC obtained from patients with FVIII inhibitors, after short-term in vitro re-stimulation of PBMC with FVIII. The FVIII-stimulated pro-inflammatory signature was not found in PBMCs obtained from patients after successful ITI, from patients without FVIII inhibitors or from healthy individuals [15]. The findings by Hofbauer et al. and Karim et al. suggest that the analysis of FVIII-induced gene expression signatures in PBMC obtained from patients exposed to FVIII replacement therapies or ITI might provide novel insights into underlying immune mechanisms regulating the development of the different populations of FVIII-specific antibodies and the eradication of FVIII inhibitors by ITI. Such analysis can be performed in small blood volumes (e.g. 3–4 mL) and can be adapted to prospective, international multi-center clinical trials. 

Clinical research in rare diseases such as Hemophilia A, especially when focused on previously untreated pediatric patients, is limited by the low incidence of patients, requiring clinical trials to be setup in a multi-center and international design. Inter-center variability is a key concern of such trials, especially if more sophisticated sample collection and processing is required. Therefore, we decided to use a model antigen (CMV-pp65) to establish suitable technologies, work-flows, standard operating procedures (SOP) and quality assurance programs to qualify local operators in Hemophilia Treatment Centers (HTC) for all the critical steps which were to be implemented in the assessment of FVIII-specific transcriptome signatures in PBMCs after short-term in vitro re-stimulation during the HIPS multicenter clinical trial. 

The outcome of our study demonstrates that HTCs are capable of performing onsite PBMC isolation and subsequent short-term in vitro re-stimulation with a model antigen, required to assess protein-specific cellular immune responses in small volumes of blood for a global, multi-center clinical trial.

## 2. Materials and Methods

### 2.1. Workflow for Qualification of Local Operators in HTC Associated Qualifying Laboratories

The workflow for the qualification of local operators in HTC associated qualifying laboratories was designed to cover all the critical steps which were to be implemented in the assessment of FVIII-specific transcriptome signatures in PBMCs after short-term in vitro re-stimulation during the HIPS multicenter clinical trial. Importantly, 4 mL cell preparation tubes (BD Vacutainer^®^ CPT™ Cell Preparation Tube with Sodium Citrate, Becton Dickinson, Franklin Lakes, NJ, USA) were used for blood collection to mimic the small blood volumes available from pediatric patients during the HIPS trial. 

The qualification procedure was implemented prior to enrollment of subjects into the HIPS study, for a minimum of 2 operators at each laboratory site. Whenever there was a change in laboratory personnel, each new operator was required to qualify prior to processing patient samples for the HIPS study. HTCs which were located within 4 h ground transportation to another HTC with an already qualified laboratory were permitted to use one of the 15 qualified HTC laboratories to process their blood samples (see Table 1). A model antigen, CMV pp65, was used for the qualification procedure to study antigen-specific re-stimulation of T-cells in PBMC obtained from anti-CMV seropositive heathy volunteer blood donors. This set-up enabled us to use blood from adult healthy blood donors in the qualification procedure. 

The final workflow of the qualification procedure is outlined in Figure 1. Two reference laboratories managed the qualification procedures for the HTC laboratories in the US and in Europe. The University of Texas Health Science Center at Houston was the reference laboratory for US HTCs. The Immunology laboratory of the Drug Discovery Austria, Baxalta Innovations GmbH, a Takeda company, Vienna, Austria, was the reference laboratory for the HTCs in Europe and acted as central laboratory for all activities related to RNA preparation, RNA quality control, cDNA synthesis and gene expression studies using Real Time qPCR. The European reference laboratory also provided all the required reagents and consumables to ensure that all qualifying laboratories used the same batches for reagents and consumables as the reference laboratories.

### 2.2. Healthy Blood Donors

Healthy volunteer blood donors (age >18 years) were selected by the 2 reference laboratories in the US and Europe. After providing informed consent, blood donors were screened for circulating IgG antibodies to CMV using a commercial enzyme-linked immunosorbent assay (ELISA) according to manufacturer’s instructions (Genway, San Diego, CA, USA and Medac, Wedel, Germany). Blood donors who were seropositive for IgG anti-CMV antibodies were eligible to donate blood for the qualification procedure.

The studies with blood obtained from healthy blood donors were approved by the IRB at the HIPS coordinating center, the (UTHealth), Houston, US, and by the ethical committee of the city of Vienna, Austria. 

### 2.3. Preparation of PBMC from Peripheral Blood and In Vitro Re-Stimulation

Sodium citrate cell preparation tubes (BD Vacutainer^®^ CPT™ Cell Preparation Tube with Sodium Citrate, Becton Dickinson, Franklin Lakes, NJ, USA) with 4 mL capacity were used for blood draw and PBMC preparation and 4 × 4 mL of blood were collected from each blood donor. The BD Vacutainer^®^ CPT™ Cell Preparation Tube with Sodium Citrate combines a blood collection tube containing a sodium citrate anticoagulant with a FICOLL™ Hypaque™ density fluid and a polyester gel barrier which separates the two liquids. This configuration permits cell separation during a single centrifugation step. The separated sample can be transported without being removed from the BD Vacutainer^®^ CPT™ Tube, since the gel forms a stable barrier between the cell layers. All procedures were carried out according to the instructions for use provided by the manufacturer Becton Dickinson. 

Blood was drawn at local medical sites which were close to the European and the US reference laboratories, respectively. The 4 CPTs containing freshly drawn blood were centrifuged once to separate mononuclear cells from granulocytes and red blood cells, according to the manufacturer’s instructions. Two of the 4 centrifuged CPTs were shipped overnight at room temperature (RT) via controlled ambient shipping containers to the qualifying HTC; the other 2 tubes remained at RT at the respective reference laboratory. Temperature during transport was monitored using a temperature logger. The next day, both the respective reference laboratory and the qualifying HTC isolated PBMC and set up the in vitro re-stimulation cultures. For this purpose, PBMC were removed from the CPT, washed twice with PBS and subsequently resuspended in 850 µL of cell culture medium, ImmunoSpot, Shaker Heights, OH, USA, supplemented with L-Glutamine, Sigma Aldrich, Darmstadt, Germany, and Penicillin/Streptomycin, ThermoFisher/GIBCO Scientific, Grand Island, NY, USA. 800 µL of this cell suspension was transferred into a Corning Costar 96 well PP cell culture plate, Kennebunk, ME, USA, (100 µL per well) and kept at 37 °C/5%CO_2_ overnight. Wells containing resting cells were surrounded by wells containing PBS, ThermoFisher Scientific, Paisley, UK, only. The remaining cell suspension was used for cell counting and assessment of cell viability using a Neubauer cell chamber, NanoEnTek, Waltham, MA, USA, and Trypan blue, Sigma Aldrich, Darmstadt, Germany, staining. On the next morning, rested cells were incubated with one of the following stimuli for 6 h: PepTivator^TM^ CMV-pp65, Milteny Biotec, Bergisch Gladbach, Germany (final concentration: 2 μg/mL, triplicates, antigen-specific re-stimulation), Dynabeads Human T-Activator CD3/CD28, ThermoFisher Scientific, Vilnius, Lithuania (final concentration: 4 × 10^5^ beads/well, duplicates, positive control) or medium only (triplicates, negative control). After 6 h incubation at 37 °C/5%CO_2_, each sample was harvested, transferred into a cryovial prefilled with RNAprotect^®^ Cell Reagent, QIAGEN, Hilden, Germany, and subsequently stored at −20 °C prior to shipment to the central laboratory. 

### 2.4. RNA Preparation and RNA Quality Control at the Central Laboratory

Cryovials containing PBMC re-stimulation mixtures transferred into RNAprotect^®^ Cell Reagent (QIAGEN, Hilden, Germany) were thawed, and the whole content was transferred into centrifugation tubes and centrifuged. Supernatants were removed and pellets were used for RNA preparation using a RNeasy Plus Micro Kit (QIAGEN, Hilden, Germany), which is suitable for small numbers of cells (≤5 × 10^5^). As a precaution, the RNeasy lysis buffer RLT was supplemented with dithiothreitol for RNase deactivation. The resulting RNA preparations were either immediately used for RNA quality control and cDNA synthesis or frozen at −80 °C until further processing. Cryovials obtained from the qualifying HTCs and from the respective reference laboratories were processed in parallel. 

RNA quality control was done using the RNA 6000 Nano Kit, Agilent, Waldbronn, Germany, for analysis on the Bioanalyzer 2100, Agilent, Singapore, Singapore, with the “Eukaryote Total RNA Nano Series II” setting, according to the manufacturer’s instructions. Results including the RNA concentration (ng/µL) and the RNA integrity number (RIN) were recorded.

### 2.5. cDNA Synthesis and Real-Time qPCR

Reverse transcription was performed with 6 µL undiluted RNA sample in a total reaction volume for cDNA synthesis of 20 µL using the Maxima Enzyme Kit (Thermo Scientific, Vilnius, Lithuania, according to the manufacturer’s instructions. Water was included as a non-template control (NTC). The Mastercycler ep Gradient S (Eppendorf, Hamburg, Germany) was utilized with the following settings: 10 min at +25 °C, 15 min at 50 °C, 5 min at +85 °C—reaction stop at +4 °C. The synthesized cDNA was either stored at −20 °C or immediately used for real-time qPCR using the ABI 7500 system (Applied Biosystems/Thermo Fisher, Carlsbad, CA, USA). 

The qPCR was carried out with a final reaction volume of 20 µL: 10 µL FAST TaqMan Universal PCR Master Mix (Qiagen, Hilden, Germany), 1 µL TaqMan FAM probes (Applied Biosystems/Thermo Fisher, Bleiswijk, The Netherlands), 9 µL pre-diluted cDNA sample. 

Cycling conditions were set to the following parameters: initial 20 s at 95 °C, followed by 40 cycles of 3 s each, denaturation at 95 °C, 30 s of annealing/extension at 65 °C followed by the hold at +4 °C. NTCs, RNA controls and interpolate controls (IPC) were included as assay controls.

Relative gene expression (2^−∆∆Ct^) of the gene encoding interferon gamma (IFNG; assay ID Hs99999041_m1) in the positive control and in CMV-pp65 stimulated samples was calculated. We chose the expression of the gene encoding IFN-γ as a read-out because it is well established that CMV-pp65- specific T cells predominantly produce inflammatory cytokines, such as IFN-γ, IL-2, and TNF-a [16]. The reference genes encoding glucuronidase beta (GUSB; assay ID Hs99999908_m1) or tyrosine 3-monooxygenase/tryptophan 5-monooxygenase activation protein zeta (YWHAZ; assay ID Hs00237047_m1), respectively, were used for normalization of gene expression data. All mentioned genes were analyzed via TaqMan FAM-MGB probes (Applied Biosystems^®^/Thermo Fisher, Bleiswijk, The Netherlands).

### 2.6. Assessment of Results and Follow-On Activities

PBMC cell count, PBMC cell viability, RIN for the assessment of RNA quality and the relative gene expression (2^−∆∆Ct^) of IFNG in PBMC stimulated with CMV-pp65 and in PBMC stimulated with Dynabeads Human T-Activator CD3/CD28 (positive control) were calculated for each blood sample. The operator in the qualifying HTC was considered qualified when all of the following criteria were met: (A)PBMC viability: we used a cutoff of ≥89% viable PBMC as previously suggested by Smith et al. [17] for effective separation of optimal PBMC samples from those that could not respond effectively to antigen.(B)PBMC yield: we requested a yield of ≥1 × 10^6^/mL PBMC as this was the minimum yield required for the subsequent gene expression studies(C)Relative IFNG gene expression: the ratio for the relative IFNG gene expression in samples processed by the operator in the qualifying HTC laboratory over the gene expression in the corresponding samples processed by the operator in the respective reference laboratory had to be within pre-defined ranges (0.67–1.5 for PBMC stimulated with CMV-pp65 and 0.8–1.25 for PBMC stimulated with Dynabeads Human T-Activator CD3/CD28). These ranges were calculated by assessing the variability for relative gene expression when three different operators in the reference laboratories processed blood of six different healthy blood donors using the workflow implemented for the qualification of local operators in HTCs, as described above.

If the qualification criteria were not achieved by the operator in the qualifying HTC, SOPs were reviewed and discussed with the operator. If required, an on-site re-training was organized. Once the potential issues were resolved, the operator was permitted to repeat the qualification procedure. If the operator passed the second attempt, this person was considered qualified.

## 3. Results

### 3.1. Qualification of Local Operators in HTC Associated Qualifying Laboratories

The qualification of local operators in HTC associated qualifying laboratories included all critical steps which were to be implemented in the assessment of FVIII-specific transcriptome signatures in PBMCs after short-term in vitro re-stimulation with FVIII during the HIPS clinical trial, as outlined in Figure 1. Temperature monitoring during overnight storage and transportation of Vacutainer Cell Preparation Tubes (CPT) used for blood draw and PBMC preparation did not show unacceptable fluctuations for any of the qualifying sites (see examples shown in Figure 2).

Thirty-nine operators at the 15 HTC associated qualifying laboratories were qualified. Thirty-one operators (79.5 %) at 11 of the qualifying HTCs had acceptable results on first testing. Eight operators from 4 HTCs failed qualification at first attempt because they had PBMC viability, PBMC count or IFNG gene expression ratios outside acceptable ranges (Table 2). All of the eight operators who failed the first qualification were successful on the second qualification attempt after a review of procedures and re-training (Table 2).

### 3.2. PBMC Viability and Yield

Both reference laboratories and qualifying operators in HTC laboratories isolated PBMC from CPT containing blood drawn from the same blood donors. PBMC yield and viability obtained by the qualifying operator were compared with the results obtained by the respective operator in the reference laboratory. The cut-off for PBMC viability was ≥89% viable cells, as previously suggested by Smith et al. [17] for effective separation of optimal PBMC samples from those that could not respond effectively to antigen (Table 3). The cut-off for PBMC yield was ≥1 × 10^6^/mL PBMC, as this was considered to be the minimum yield required for the intended gene expression studies (Table 3). 

Acceptable results for PBMC yield on first attempt were obtained by 32 of the 39 operators in the qualifying HTC laboratories. Seven operators were retrained and achieved acceptable results on the second attempt (Table 2). The reasons for failure were low PBMC yields caused by suboptimal recovery of PBMC from CPT, wrong dilution of cells prior to cell counting, or failure in calculating cell concentrations based on the counting result. Thirty-seven of the 39 operators in the qualifying HTC laboratories achieved acceptable results for PBMC viability on the first attempt. Two operators failed the first qualification due to a faulty assessment of cell yield. They were retained and achieved acceptable results on the second attempt (Table 2).

### 3.3. Gene Expression Data

All gene expression studies including RNA preparation, cDNA synthesis and real-time qPCR were performed in the central laboratory in Vienna, Austria. The acceptance criteria for passing the qualification test were based on the pre-defined ratio for the relative IFNG gene expression in samples generated by the operator in the qualifying HTC laboratories over the gene expression in the corresponding samples generated by the operator in the respective reference laboratory. Three of the 39 qualifying operators failed these acceptance criteria for gene expression ratios at first attempt, but passed after retraining at the second attempt (Table 2). Examples for IFNG gene expression ratios obtained by operators 1, 2 and 3 who passed on first qualification attempt and of operators 32, 33 and 34 who failed these acceptance criteria on first attempt are provided in Figure 3 and Figure 4.

### 3.4. RNA Yield and RNA Quality

RNA preparation and RNA quality assessment for all samples generated in the qualifying HTCs or in the respective reference laboratories were carried out in the central laboratory in Vienna, Austria. There was good agreement for concentration and RNA Integrity Number (RIN) of RNA samples, isolated from PBMC after in vitro re-stimulation with CMV pp65 or positive/negative controls, between samples generated by HTC site operators who passed the qualification and corresponding samples generated by operators in the respective reference laboratories (Table 4).

## 4. Discussion

Establishing training and qualification test procedures for local operators in clinical HTCs is essential to enable reliable and reproducible gene expression studies in PBMC obtained from small blood samples from very young children during multicenter clinical trials. The level of experience and skill of laboratory staff performing PBMC isolation is a major determinant of the viability and functionality of PBMCs [18]. Therefore, qualification of operators for PBMC preparation during clinical trials is crucial [19,20,21].

The most important aspects of the qualification test procedures presented in this manuscript were the establishment of SOPs, the training of all participating operators in these SOPs and the on-site training and qualification. The qualification of operators included qualification runs which covered all on-site laboratory activities necessary to study FVIII-induced gene expression signatures in previously untreated hemophilia A patients during the prospective multi-center HIPS trial. 

We used BD Vacutainer CPT for blood draw and preparation of PBMC. Previous studies indicated that CPT are suitable for collecting high-quality PBMC required for downstream immunological assays [22,23,24,25]. Compared to the use of high-density medium alone, e.g., Ficoll-Paque^®^, CPT have simplified the processing of PBMC from blood by eliminating a dilution and overlay step prior to centrifugation. Moreover, CPT facilitate shipping of centrifuged vacutainers for further off-site processing [25]. These features are particularly relevant in multi-center studies that are limited to the small blood volumes obtainable from infants. 

Another important consideration was the choice of a suitable model antigen for the set-up of the qualification test procedures. We selected a peptide pool consisting mainly of 15-mer peptides covering the complete sequence of the CMV-pp65 protein. CMV-pp65 is an immunodominant target of CD4+ and CD8+ T-cell responses to CMV. It is well established that CMV-pp65- specific T cells predominantly produce inflammatory cytokines, such as IFN-γ, IL-2, and TNF-α [16,26,27]. Moreover, there is a good correlation between CMV seropositivity and the presence of CMV-pp65-specific T cells in the circulation of healthy blood donors [16]. These features were highly relevant for us because they enabled us to work with blood samples from healthy adults who could be easily tested for CMV seropositivity. 

Immunological studies of rare disorders such as severe hemophilia A frequently require multi-center and international collaboration to meet recruitment goals. Not all HTCs with the clinical expertise to participate in the HIPS study had laboratory facilities which could support preparation of PBMC from circulating blood, the set-up of a short in vitro re-stimulation of PBMC and the subsequent transfer of the re-stimulation mixture into RNAprotect for storage. Therefore, we included regional laboratories which were available within a maximum of 4 h of ground transportation in the qualification procedures. In earlier studies, we had demonstrated that the temperature-monitored transport of CPT containing freshly drawn blood did not negatively impact the collection of high-quality PBMC required for downstream gene expression analysis after short in vitro re-stimulation with a model antigen. This way, the HIPS study network of participating HTCs could be expanded to an additional six sites.

Our data indicate that, considering all the required tasks, 31 qualifying operators at 11 HTCs passed the qualification per HTQ criteria on the first attempt. In total, eight qualifying operators from 4 HTCs failed to qualify at the first attempt. Four operators failed the acceptance criterion for PBMC yield, two operators failed the acceptance criteria for both PBMC yield and IFNG gene expression, one operator failed the acceptance criterion for PBMC cell viability and one operator failed the acceptance criteria for all three parameters. However, after a review of the SOPs and, if required, an on-site re-training, these eight operators passed the qualification on the second attempt. These results highlight that the development of SOPs and the on-site training of HTC operators on these SOPs are essential components to enable multicenter studies of gene expression signatures. None of the HTCs had any identified failure in yield or viability of PBMC after their operators had passed the initial qualification. 

The establishment of quality assurance procedures has been recognized as essential for obtaining reliable laboratory data in a multicenter clinical study. In 2006, Dyer et al. described the establishment of a multicenter quality assurance program in clinical laboratories across Australia to evaluate PBMC quality (yield, viability and function) and to support participating laboratories in improving their performance [20]. In 2014, the HIV Vaccine Trials Network, a global network of 28 clinical trial sites, reported the implementation of a comprehensive PBMC quality program that included staff training and quality management systems that would mitigate risk to successful vaccine development [19]. These are only two examples out of a range of PBMC quality programs which were recently described. The findings during the reported implementation of the PBMC quality programs confirm our own data, indicating that the harmonization of operating procedures by the establishment of SOPs and the qualification of local operators can significantly improve performance and enable high-quality multicenter clinical studies requiring functional analysis of PBMC. 

## 5. Conclusions

In conclusion, our data indicate that the establishment of SOPs with on-site training of HTC operators and qualification test procedures are necessary and feasible to ensure reproducible, high-quality sample collection and processing in multicenter clinical studies of gene expression signatures in human PBMC obtained from small blood volumes.

## Figures and Tables

**Figure 1 jcm-12-02080-f001:**
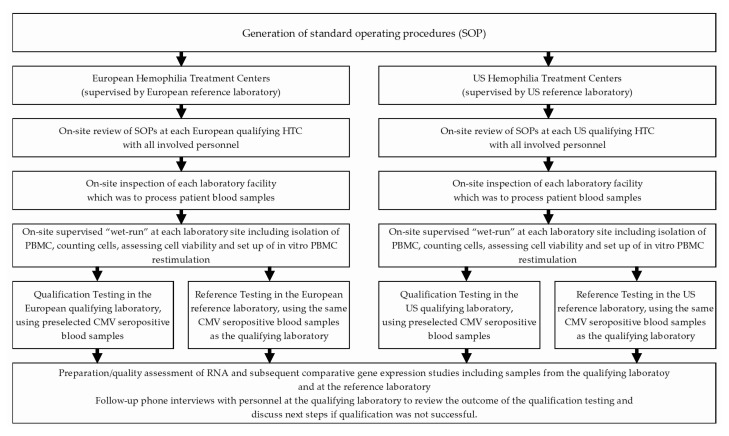
Workflow for qualification of local operators in Hemophilia Treatment Centers (HTCs).

**Figure 2 jcm-12-02080-f002:**
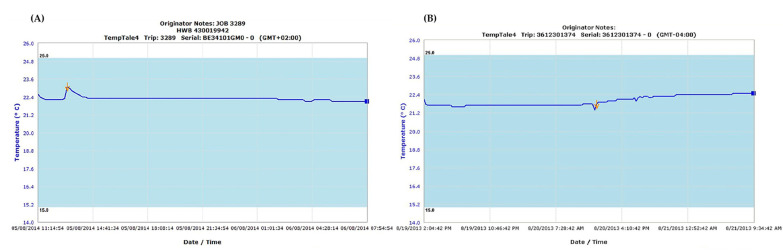
Two representative examples for temperature monitoring during shipment of CPT tubes containing blood draws. (**A**) Results for temperature monitoring recorded during the transport of CPT tubes containing blood draws from the European reference lab to a European qualifying laboratory site. (**B**) Results for temperature monitoring recorded during the transport of CPT tubes containing blood draws from the US reference lab to a US qualifying laboratory site.

**Figure 3 jcm-12-02080-f003:**
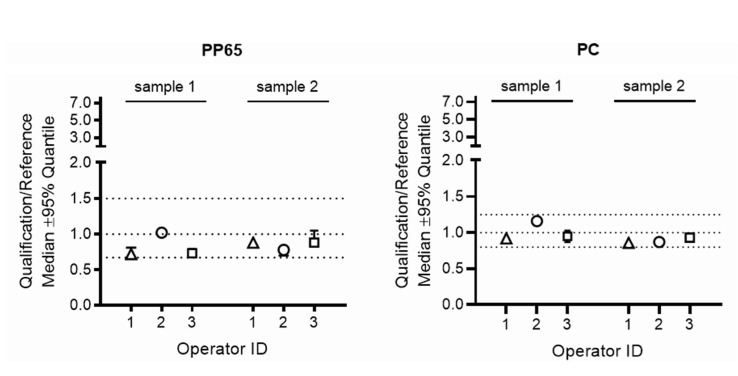
IFNG gene expression ratios obtained by operators who passed at first qualification attempt. Presented are representative examples for IFNG gene expression ratios obtained by qualifying operators (operator ID 1, 2, 3) who passed at first qualification attempt. The predefined ranges of acceptance for the ratios of the relative IFNG gene expression in samples processed by the operator in the qualifying HTC laboratory over the gene expression in the corresponding samples processed by the operator in the respective reference laboratory are marked by dotted lines, 0.67–1.5 for PBMC stimulated with CMV-pp65 (PP65) and 0.8–1.25 for PBMC stimulated with Dynabeads Human T-Activator CD3/CD28 (PC). PBMC sample 1 and PBMC sample 2 were prepared from the blood of two preselected healthy blood donors who were positively tested for anti-CMV IgG antibodies. The ratio for the relative IFNG gene expression in samples processed by the operator in the qualifying HTC laboratory over the gene expression in the corresponding samples processed by the operator in the respective reference laboratory were calculated. PP65: Results obtained from PBMC stimulated with CMV PP65. All tests for CMV PP65-stimulated cultures and for negative control cultures were run in triplicate. PC: Results obtained from PBMC stimulated with the positive control (Dynabeads Human T-Activator CD3/CD28). All tests for positive control cultures were run in duplicates.

**Figure 4 jcm-12-02080-f004:**
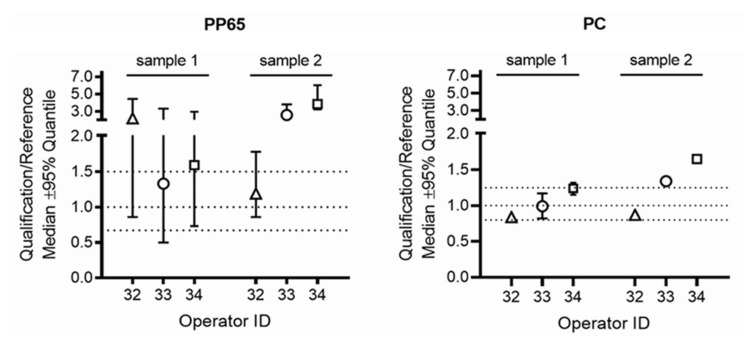
IFNG gene expression ratios obtained by operators who failed at first qualification attempt. Presented are representative examples for IFNG gene expression ratios obtained by qualifying operators (operator ID 32, 33, 34) who failed at the first qualification attempt. The predefined ranges of acceptance for the ratios of the relative IFNG gene expression in samples processed by the operator in the qualifying HTC laboratory over the gene expression in the corresponding samples processed by the operator in the respective reference laboratory are marked by dotted lines, 0.67–1.5 for PBMC stimulated with CMV-pp65 (PP65) and 0.8–1.25 for PBMC stimulated with Dynabeads Human T-Activator CD3/CD28 (PC). PBMC sample 1 and PBMC sample 2 were prepared from the blood of two preselected healthy blood donors who were positively tested for anti-CMV IgG antibodies. The ratio for the relative IFNG gene expression in samples processed by the operator in the qualifying HTC laboratory over the gene expression in the corresponding samples processed by the operator in the respective reference laboratory was calculated. PP65: Results obtained from PBMC stimulated with CMV PP65. All tests for CMV PP65-stimulated cultures and for negative control cultures were run in triplicate. PC: Results obtained from PBMC stimulated with the positive control (Dynabeads Human T-Activator CD3/CD28). All tests for positive control cultures were run in duplicates.

**Table 1 jcm-12-02080-t001:** Hemophilia Treatment Centers, Associated Qualifying Laboratories and responsible Reference Laboratories.

Hemophilia Treatment Center	Associated Qualifying Laboratory	Reference Laboratory
University Texas, Houston, TX, USA	University Texas, Houston, TX, USA	University Texas, Houston, TX, USA
Baylor College of Medicine, Houston, TX, USA	University Texas, Houston, TX, USA	University Texas, Houston, TX, USA
Cincinnati Children’s Hospital, Cincinnati, OH, USA	Cincinnati Children’s Hospital, Cincinnati, OH, USA	University Texas, Houston, TX, USA
Emory University, Atlanta, GA, USA	Emory University, Atlanta, GA, USA	University Texas, Houston, TX, USA
Indiana Hemophilia and Thrombosis Center, Indianapolis, IN, USA	Cincinnati Children’s Hospital, Cincinnati, OH, USA	University Texas, Houston, TX, USA
Miami University, Miami, FL, USA	Miami University, Miami, FL, USA	University Texas, Houston, TX, USA
Oregon Health & Science University, Portland, OR, USA	Oregon Health & Science University, Portland, OR, USA	University Texas, Houston, TX, USA
Tulane University, New Orleans, LA, USA	Tulane University, New Orleans, LA, USA	University Texas, Houston, TX, USA
University of Iowa, Iowa City, IA, USA	University of Iowa, Iowa City, IA, USA	University Texas, Houston, TX, USA
University of Kentucky, Lexington, KY	Cincinnati Children’s Hospital, Cincinnati, OH, USA	University Texas, Houston, TX, USA
University North Carolina, Chapel Hill, NC, USA	University North Carolina, Chapel Hill, NC, USA	University Texas, Houston, TX, USA, USA
University of Utah, Salt Lake City, UT, USA	University of Utah, Salt Lake City, UT, USA	University Texas, Houston, TX, USA
Weill Cornell Medicine, New York City, NY, USA	Weill Cornell Medicine, New York City, NY, USA	University Texas, Houston, TX, USA
University Pittsburgh Medical Center, Pittsburgh, PA, USA	University Pittsburgh Medical Center, Pittsburgh, PA, USA	University Texas, Houston, TX, USA
North Texas Comprehensive Hemophilia Center, Children’s Medical Center, Dallas, TX, USA	North Texas Comprehensive Hemophilia Center, Children’s Medical Center, Dallas, TX, USA	University Texas, Houston, TX, USA
University of Amsterdam, Amsterdam, The Netherlands	Sanquin Research, Amsterdam, The Netherlands	Baxalta Innovations GmbH, Vienna, Austria
Lund University, Malmö, Sweden	Lund University, Malmö, Sweden	Baxalta Innovations GmbH, Vienna, Austria
University of Milan, Milan, Italy	University of Milan, Milan, Italy	Baxalta Innovations GmbH, Vienna, Austria
Masaryk University, Brno, Czech Republic	Baxalta Innovations GmbH, Vienna, Austria	Baxalta Innovations GmbH, Vienna, Austria
Medical University Vienna, Vienna, Austria	Baxalta Innovations GmbH, Vienna, Austria	Baxalta Innovations GmbH, Vienna, Austria

**Table 2 jcm-12-02080-t002:** Qualifying operators who failed the first qualification and had to be retrained.

Operator ID	Retraining/Failed Qualification	Reason for Failure	Resolution
8, 9, 24, 25, 32, 33, 34	PBMC yield	Low PBMC yields caused by suboptimal recovery of PBMC from CPT tubes, wrong dilution of cells prior to cell counting or failure in calculating cell concentrations based on the counting result	Retraining on SOPs, all operators passed on second attempt
14, 32	PBMC cell viability	Faulty viability assessment	Retraining on SOPs, all operators passed on second attempt
32, 33, 34	IFNG gene expression	IFNG gene expression ratios compared to reference lab were not acceptable	Retraining on SOPs, all operators passed on second attempt

PBMC: Peripheral blood mononuclear cells; CPT tubes: cell preparation tubes. SOP: Standard operating procedure; IFNG: Interferon gamma.

**Table 3 jcm-12-02080-t003:** PBMC viability and PBMC yield obtained by qualifying HTC operators, compared to the results of the operators in the respective Reference lab.

	PASS		FAIL	
	HTC Operators	Reference Lab	HTC Operators	Reference Lab
PBMC viability [%]	97.7	98.0	60.0	97.0
Median [min, max]	[89.0, 100.0]	[92.0, 100.0]	[52.0, 66.0]	[97.0, 98.0]
PBMC yield [×10^6^/mL]	4.8	5.0	0.2	3.6
Median [min, max]	[2.0, 10.0]	[1.3, 12.5]	[0.1, 0.8]	[2.2, 7.0]

PBMC: Peripheral blood mononuclear cells. PASS: HTC operators who passed the qualification. FAIL: HTC operators who failed the qualification.

**Table 4 jcm-12-02080-t004:** Assessment of RNA samples isolated from PBMC after in vitro re-stimulation with CMV pp65 or negative/positive controls.

PBMC Stimulus	RNA Conc [ng/mL]		RIN	
	HTC Operators	Reference Lab	HTC Operators	Reference Lab
CMV pp65	13.0 [1.1, 37]	12.0 [6.0, 32.0]	9.4 [8.1, 10.0]	9.3 [8.4, 10.0]
*n*	39	39	35	36
Negative control	15.0 [1.0, 63.0]	12.0 [0.2, 40.0]	9.4 [8.0, 10.0]	9.5 [7.6, 10.0]
*n*	39	39	35	36
Positive control	14.0 [1.0, 52.0]	11.0 [1.1, 41.0]	9.5 [7.7, 10.0]	9.4 [8.0, 10.0]
*n*	39	39	33	33

Presented are median (min, max) values for the RNA concentration (RNA conc.) and the RNA Integrity Number (RIN) of samples isolated from in vitro re-stimulated PBMC, prepared by either Qualifying operators who had passed the qualification (HTC operators) or by operators in the respective reference labs. *n*: number of samples (number of reference lab values are counted for each operator being compared for the same blood donor samples).

## Data Availability

The datasets used and/or analysed during the current study are available from the corresponding author on reasonable request.

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
