# Peer review of "Qualification of Hemophilia Treatment Centers to Enable Multi-Center Studies of Gene Expression Signatures in Blood Cells from Pediatric Patients"

_jcm, 2023, doi:10.3390/jcm12052080_

Round 1
Reviewer 1 Report
In this manuscript, in order to monitor longitudinal FVIII-specific gene expression signatures in pediatric HA patients with inhibitors following FVIII exposure from different medical centers, authors aimed at establishing SOPs and qualification test procedures from specimen collection, transport, storage, PBMC isolation, RNA preparation, and subsequent RT-qPCR using a model antigen CMV-pp65 antigen. the results showed that the goal is achievable through on-site trainings and quality assurance. Overall, the study was well designed and the data were convincing. However, there are some concerns.
1. The manuscript mainly focus the establishment of procedures and test effectiveness of procedures using a model antigen. But the introduction section talked too much about FVIII inhibitors and Hemophilia, FVIII exposure, and related literatures, etc. it needs to be simplified and be concise.
2. In the end of the introduction section, authors stated 'the outcome of our study demonstrates that HTCs are capable of performing onsite PBMC isolation and subsequent short-term in vitro re-stimulation with FVIII, required to assess FVIII-specific cellular immune responses in small volumes of blood for a global, multi-center clinical trial.' However, the data provided in this manuscript was nothing to do with FVIII re-stimulation, instead, only with a model antigen. it is over stated and need to be rephrased.
3. In the 'Materials and Methods' section, authors didn't specified how to centrifuge CPTs containing freshly drawn blood (lack of information about centrifuge speed and time, type of centrifuge). And if it is centrifuged prior to transport, it is possible that the whole blood mixed again during transport. So what is the point to centrifuge them prior to transport? or if the centrifuge is necessary, how to make sure that blood would not to remix during transport?
Reviewer 2 Report
Minor Comments.
Page 1, lines 5-6 and 19-28.
The way how the abstract is written makes impression that the Hemophilia Treatment Centers deal with only abnormalities due to development of anti-FVIII antibodies in humans. In particular, such is immediately felt upon the reading of the title and abstract. I suggest modifying the abstract (or/and title) to achieve a balance between the center naming and first part of the abstract text which are the most discrepant.
Page 2, line 55.
It is recommended to state why the development of anti-FVIII non-neutralizing antibodies is harmful for hemostasis. In particular, upon the binding, the plasma clearance of such complexes is likely faster that decreases the efficacy of FVIII replacement therapy. Respective reference(s) should be provided.
Page 2-3, lines 99-100
“In the course of HIPS study”. There is no need in the word “study”, as it was already meant under HIPS.
Page 3, line 130
Is should be spelled out the abbreviation of “CMV pp65” and explained why it was chosen as a model antigen.
Page 5, line 176.
Please correct the formatting (left margin).
Page 6, line 194-195.
Amounts of the stimulation compounds should be provided. In particular, such is important for CMV pp65 mimicking stimulation by FVIII in real experiments.
Page 6, line 223.
(i) Please provide a short description of oligo primers (gene specificity) used for qPCR (is it TaqMan FAM probes?). Not all readers understand this. (ii) Please spell out the genes IFN-γ, GUSB and YWAHZ (products) names and briefly explain why two latter ones were used.
Major Comment.
Through the whole paper, it is assumed that if an operator is successful in mastering the assay with the model antigen CMV pp65, instead FVIII, the assay will work in his/her hands. However, such assumption may be incorrect as FVIII is a “tricky” protein. Therefore, I believe that the training protocol should be validated by inclusion of the arm with FVIII antigen. If it is demonstrated that the FVIII-specific assay performs equally with CMV ppp65-specific assay in hands of several (5-7) operators, then such approach will be considered validated, and the protocol would be justified. However, if the authors already performed such validation in their previous work, this needs to be clearly stated (in Introduction) and mentioned in Discussion.
In any case, this matter should be substantially discussed in the Discussion (page 11).
Reviewer 3 Report
The authors describe a training and qualification programme for 39 operators from 15 clinical sites to enable PBMC isolation and gene expression studies to be performed from a variety of geographical locations. This is important as standardization and validation of testing at different locations allow for recruitment of higher numbers of patients, important in rare diseases such as haemophilia. A particular strength of the study is the emphasis on small volume samples of importance in studies involving very young children. The manuscript is well-written with detailed technical descriptions and references. I have 2 minor comments for the authors to address:
1. Following the initial qualification, will there be regular quality assessments and at what intervals?
2. Are there plans to expand the number of reference laboratories and qualified centres beyond Europe and North America so that additional information from different ethnic groups are captured?
Round 2
Reviewer 2 Report
Thank you for addressing my previous comments and explanations.